# Impact of Mineralocorticoid Receptor Gene *NR3C2* on the Prediction of Functional Classification of Left Ventricular Remodeling and Arrhythmia after Acute Myocardial Infarction

**DOI:** 10.3390/ijerph20010012

**Published:** 2022-12-20

**Authors:** Rima Braukyliene, Ali Aldujeli, Ayman Haq, Laurynas Maciulevicius, Darija Jankauskaite, Martynas Jurenas, Ramunas Unikas, Vytautas Zabiela, Vaiva Lesauskaite, Sandrita Simonyte, Diana Zaliaduonytė

**Affiliations:** 1Department of Cardiology, Faculty of Medicine, Medical Academy, Lithuanian University of Health Sciences, A. Mickeviciaus 9, 44307 Kaunas, Lithuania; 2Kaunas Region Lithuanian Society of Cardiology, Eiveniu Str. 2, 50009 Kaunas, Lithuania; 3Laboratory of Molecular Cardiology, Lithuanian University of Health Sciences, Sukileliu 15, 50162 Kaunas, Lithuania; 4Minneapolis Heart Institute, 800 E 28th St Heart Hospital Minneapolis, Minneapolis, MN 55407, USA

**Keywords:** NR3C2 gene, acute myocardial infarction, mineralocorticoid receptor, rs5522, rs4635799, rs2070950

## Abstract

Background: The NR3C2 gene encodes the mineralocorticoid receptor, which is present on cardiomyocytes. Prior studies reported an association between the presence of NR3C2 single-nucleotide polymorphisms (SNPs) and an increased cortisol production during a stress response such as acute myocardial infarction (AMI), which may lead to adverse cardiac remodeling. Objective: To study the impact of the NR3C2 rs2070950, rs4635799 and rs5522 gene polymorphisms on left ventricular (LV) remodeling, rhythm and conduction disorders in AMI patients. Methods: A cohort of 301 AMI patients who underwent revascularization was included. SNPs of the NR3C2 gene (rs2070950, rs4635799 and rs5522) were evaluated. A total of 127 AMI patients underwent transthoracic echocardiography follow-up after 72 h and 6 months. Results: The rs2070950 GG genotype and rs4635799 TT genotype were most common in patients who had LV end-diastolic volume increase < 20% and the same or increased LV ejection fraction, indicating a possible protective effect of these SNPs. The rs5522 TT genotype was associated with a higher frequency of arrhythmias, while the presence of at least one rs5522 C allele was associated with a lower risk of arrhythmias. Conclusion: SNPs of the NR3C2 gene appear to correlate with better ventricular remodeling and a reduced rate of arrhythmias post-AMI, possibly by limiting the deleterious effects of cortisol on cardiomyocytes.

## 1. Introduction

Patients with acute myocardial infarction (AMI) experience both physical and psychological stress, accompanied by its physiological manifestations [1,2,3]. A cascade of events during the early phase of an AMI activates the neurohormonal system in an effort to preserve circulatory homeostasis. However, these neurohormonal changes may have deleterious effects if their activity is long enough to cause an increase in sensitivity to catecholamines and an increase in the mineralocorticoid receptors present in the myocardium [2,4].

Cortisol is the final product of the hypothalamic–pituitary–adrenal axis (HPA) and is a primary stress hormone that acts via glucocorticoid receptors (GRs) and mineralocorticoid receptors (MRs), both of which are present in cardiomyocyte nuclei [5,6]. MRs are located in the kidney, colon and salivary glands, and facilitate sodium reabsorption upon activation of the renin–angiotensin–aldosterone system. Conversely, MRs in other structures such as the brain, heart, blood vessels or adipose tissue have multiple and less clearly characterized roles. Evidence has shown that the MRs in cardiac tissue are crucial for the process of cardiac remodeling and arrhythmias [7]. A study on animal models performed in 2017 showed that the activation of MRs increases cardiomyocyte oxidative stress and inflammation, which leads to adverse cardiac tissue remodeling. This includes adverse electrophysiological remodeling, such as atrial and ventricular arrhythmias, which may be prevented by mineralocorticoid receptor antagonist (MRA) treatment [3].

Tachyarrhythmias remain a major cause of cardiac death, with an incident rate of 1.4% occurring within the first month post-AMI. This risk is related to the interplay of multiple post-AMI factors, mostly between scar tissue formation, cardiac remodeling, neurohormonal disbalances, sporadic ischemia, reduced autonomic tone and abnormalities in cardiac repolarization [8,9,10]. Currently, there are no credible tools that can guide clinicians in stratifying the patient risk of sudden cardiac death resulting from cardiac tachyarrhythmia.

The human NR3C2 gene (nuclear receptor subfamily 3 group C member 2) is a protein-coding gene that encodes MRs. It consists of ten exons and is located in chromosome 4q 31.23 [11,12]. Several studies have reported an association between the presence of MR NR3C2 polymorphism rs5522 (c.538G > A, p.Val180Ile) and an increased cortisol production after stress due to greater HPA-axis reactivity [13,14]. To the best of our knowledge, no data are currently available with regard to the study of the impact of the NR3C2 rs2070950, rs4635799 and rs5522 gene polymorphisms on post-AMI patients. The purpose of the present study was to investigate the impact of the above-mentioned gene polymorphisms on left ventricular remodeling, rhythm and conduction disorders occurring in a cohort of AMI patients who underwent primary percutaneous coronary intervention (PCI).

## 2. Material and Methods

### 2.1. Study Population

A total of 301 AMI patients who underwent primary percutaneous coronary intervention (PCI) and guideline-directed medical treatment [15] from April 2018 to November 2020 at the Hospital of Lithuanian University of Health Sciences were included in the study. Patients with previous history of acute coronary syndrome and revascularization were excluded so as to avoid including patients with previous cardiac remodeling events. However, 174 patients were lost on follow-up after 6 months.

All patients were given a written informed consent before enrollment. The study protocol conforms to the ethical guidelines of the 1975 Declaration of Helsinki as reflected in a prior approval by the Regional Biomedical Research Ethics Committee of the Lithuanian University of Health Sciences (ID No. BE-2-4).

### 2.2. Blood Samples

Venous blood samples were drawn by the standard venipuncture procedure for each study patient on admission. The samples were allowed to clot at room temperature, and the sera were separated for analyses. The serum concentrations of cortisol and troponin I were quantified by using commercially available kits (ST AIA-PACK CORT and ST AIA-PACK cTnI 3rd Gen, respectively) using the automated enzyme immunoassay analyzer AIA-2000 (Tosoh Corporation, Tokyo, Japan) while following the manufacturer recommendations. The normal value of cortisol was considered as 177–578 nmol/L at 7 a.m., and <434 nmol/L at 4 p.m.

### 2.3. Genetic Analysis

For DNA extraction, blood samples were collected from each individual in ethylenediaminetetraacetic (EDTA) tubes during their health examination. DNA was extracted from peripheral blood leukocytes by using Invitrogen DNA isolation kit, PureLink™ Genomic DNA Mini Kit (ThermoFisher Scientific, Vilnius, Lithuania), according to the manufacturer’s instructions. Single-nucleotide polymorphisms (SNPs) of NR3C2 (rs2070950, rs4635799 and rs5522) were evaluated by using real-time PCR with TaqMan^®^ SNP Genotyping Assays (C___1594391_1, C___1594397_10 and C__12007869_20, respectively) according to the manufacturer’s instructions (Applied Biosystems™, Waltham, MA, USA). The cycling program started with heating to 95 °C for 5 min, followed by 40 cycles (at 95 °C for 15 s and at 60 °C for 1 min). Allele-specific fluorescence was analyzed on the QuantStudio™ 5 Real-Time PCR System (Applied Biosystems™, USA).

### 2.4. Echocardiographic Data Acquisition

All patients underwent transthoracic echocardiography 72 h after admission and at a 6-month follow-up. Echocardiography was performed by using a Philips iU22 ultrasound machine. M-mode and 2D images were obtained and saved in cine-loop format. The LV end-diastolic volume (LVEDV), end-systolic volume (LVESV) and left ventricular ejection fraction (LVEF) were assessed by using the biplane Simpson’s disc summation method through QLAB ultrasound cardiac analysis on apical two, three and four chamber views. Four LV remodeling subgroups were identified, based on the change in LVEDV and LVEF from baseline to 6 months post-infarct. A ≥ 20% increase in LVEDV, compared to the baseline, was defined as LV dilatation, while any increase or decrease in LVEF compared to the baseline was used to categorize remodeling further according to the systolic function. Four distinct functional LV remodeling groups (FLVR) were defined as follows: Group 1—EDV < 20% and the same or increased LVEF; Group 2—EDV < 20% and a decrease in LVEF; Group 3—EDV ≥ 20% and the same or increased LVEF; group 4—EDV ≥ 20% and decreased LVEF [16].

### 2.5. Statistical Analysis

Continuous variables were expressed as means ± standard deviations (SDs) when normally distributed, and as medians and the interquartile range (IQR) when not normally distributed. The χ^2^, Kruskal–Walli’s and ANOVA tests were used to analyze differences in four FLVR groups and in the patients’ baseline characteristics. Ordinal logistic regression analysis was used for comparing the four FLVR groups: Group 1—EDV < 20% and an EF increase; Group 2—EDV < 20%, an EF decrease; Group 3—EDV ≥ 20%, an EF increase; Group 4—EDV ≥ 20%, an EF decrease. The χ^2^ test was used for the assessment of the Hardy–Weinberg equilibrium (HWE) for the distribution of genotypes. The differences between FLVR groups after MI and NR3C2 gene polymorphisms were analyzed by using Fisher’s exact test. The significance of the genetic association was measured by using the Z test.

The χ^2^ or Fisher’s exact test was used for the analysis of the rhythm, conduction disorders and NR3C2 gene polymorphisms. All of the statistical analyses were performed with SPSS 27.0 software (SDSPSS, Chicago, IL, USA). The results were considered statistically significant when the two-tailed p-value was <0.05.

## 3. Results

### 3.1. Baseline Patients’ Characteristics

In the second and fourth FLVR groups, there were significantly younger patients than in the first FLVR groups (mean age 58.4 and 57.1 years vs. 65.0 years, *p* = 0.006) (see Table 1). In all the groups, male patients constituted approximately two-thirds of the patients. Blood test parameters and risk factors of ischemic heart disease such as diabetes mellitus, hypertension, dyslipidemia and the family history of heart attacks did not differ across the four FLVR groups. However, there were more smokers in the fourth group than in the first and third FLVR groups (67.3% vs. 24.2% and 24.3%, *p* = 0.002). Moreover, the patients in the fourth FLVR group had higher glucose levels on admission than the patients representing the other FLVR groups. More than 80% of the studied population was taking dual antiplatelet therapy, ACE or ARB inhibitors, beta-blockers and statins on discharge. However, the fourth FLVR group’s patients were more frequently on MRAs at discharge when compared to the other groups.

### 3.2. Baseline Echocardiography Characteristics

When comparing the participants across the four FLVR groups, patients with a higher LVEDV_2_ and LVESV_2_ or a lower LVEF_1&2_, greater LV myocardial mass_1_, and lower mitral A wave velocity_1_ and LV E’ average_1_ were more likely to fall into a higher FLVR group (Table 2).

Group 1—EDV < 20% and the same or increased LVEF; Group 2—EDV < 20% and a decrease in LVEF; Group 3—EDV ≥ 20% and the same or increased LVEF; Group 4—EDV ≥ 20% and decreased LVEF.

### 3.3. NR3C2 Genotype Associations with Four FLVR Groups

The *NR3C2* genotype and allele frequency distribution of the fourth FLVR group were determined according to the Hardy–Weinberg equilibrium (*p* > 0.05).

It was found that rs5522 gene polymorphism was not significantly associated with the FLVR groups after MI.

Patients after MI were carrying the rs2070950 GG genotype more frequently in the first FLVR group as compared with the second (*p* < 0.05) and fourth (*p* < 0.05) groups, and in the third group more frequently than in the second group (*p* < 0.05), while no patients were carrying the GG genotype in the second and the fourth FLVR groups (Figure 1).

In addition, patients after MI had the rs2070950 C allele more frequently in the second FLVR group than in the first and third FLVR groups (*p* < 0.05) (Figure 2).

When analyzing the rs4635799 allele, more patients in the first FLVR group were carrying the TT genotype than in the second group (*p* < 0.05), and the patients in the fourth group were not carrying the TT genotype (*p* < 0.05) (Figure 3).

The C allele was detected more frequently in the second FLVR group than in the first group (*p* < 0.05) (Figure 2).

### 3.4. NR3C2 Gene Polymorphisms and Serum Cortisol Association with Rhythm and Conduction Disorders

The NR3C2 genotypes did not correlate with the serum cortisol level during the early phase of AMI. Patients with ventricular arrhythmia (VA) (ventricular tachycardia (VT), ventricular fibrillation (VF) and ventricular flutter (VFL)) and AF/AFL during the early phase of AMI had a higher serum cortisol level than the patients without rhythm disorders (*p* = 0.001) (Table 3).

The *NR3C2* genotype and allele frequency distribution of the rhythm and conduction disorders were found according to the Hardy–Weinberg equilibrium (*p* > 0.05). It was found that rs5522 gene polymorphism was significantly associated with rhythm disorders, including VT, VF/VFL and AF/AFL, during hospitalization for AMI. The rs5522 TT genotype frequency was higher in patients with AF/AFL than in patients without AF/AFL during the hospital stay (97.5% vs. 77.7%). C allele frequency was 19.6%, and a protective effect was observed for AF/AFL during the hospital stay (0.12% and 0.01% in patients without AF/AFL and with AF/AFL, respectively; *p* = 0.003). There was also a protective effect in VA (0.11% and 0.0% in patients without VA and with VA, respectively; *p* = 0.028) (Table 4).

However, the gene polymorphism was not associated with the high-grade atrioventricular block (HAVB) (Mobitz type II second-degree or third-degree AV block) [17].

On the contrary, the rs2070950 and rs4635799 gene polymorphisms were not associated with rhythm and conduction disorders (Table 4).

## 4. Discussion

In this prospective study of patients who were presented with NSTEMI or STEMI, we found that the *NR3C2* rs2070950 and rs4635799 gene polymorphisms were significantly associated with the left ventricular remodeling, and the rs5522 gene polymorphism was significantly associated with cardiac conduction disorders in the post-AMI period. This may be due to the role of MRs in both mechanical and electric cardiac remodeling.

The association between stress-induced cortisol reactivity and cognition is strongly associated with genetics. Thomas Plieger et al. studied four SNPs of the *NR3C2* gene (rs6810951, rs4635799, rs11099695 and rs2070950) coding for MRs in healthy males before and after experiencing an acute laboratory stressor when performing tasks assessing attention and reasoning (the Socially Evaluated Cold Pressor Test, SECPT). Haplotype analysis revealed that *NR3C2* had significant effects on the cortisol stress response [18].

Based on the findings of our research, the percentage of people who smoked in the fourth group was noticeably greater when compared to the percentages in the other groups. According to the findings of Leigh et al., a higher mean pack-year total smoked throughout a lifetime is associated with a higher LVM, worse LV geometry and worse diastolic function in those who are presently smoking [19].

Patients post-AMI suffer from considerable reversed left ventricular remodeling, which leads to a higher risk of developing heart failure, cardiac arrhythmias and overall worse outcomes. Therefore, the early detection of patients with a higher risk of reversed remodeling can aid clinicians in preventing poor outcomes. According to the established literature, the process of cardiac tissue remodeling starts just after an AMI. This can be due to the activation of the neurohumoral system after STEMI, which is a key player in LV remodeling [20]. Previous studies reported several variants of genes related to the neuroendocrine system that were able to predict post-AMI remodeling [21,22].

Chimed et al. described four FLVR groups and established that LV dilatation accompanied by the reduction in the LV systolic function was associated with the worst prognosis in post-AMI patients [16]. Our study showed that patients with the *NR3C2* rs2070950 GG genotype and rs4635799 TT genotype are protected for the reduced LV systolic function despite a ≥ 20% increase or a decrease in the LVED (FLVR 2nd and 4th groups) at 6-month follow-up post-AMI. However, 70% of the patients holding the *NR3C2* rs2070950 and rs4635799 C alleles fell in the 2nd FLVR group after MI. On the other hand, we failed to establish any significant association between rs5522 gene polymorphism and FLVR.

Alessandra Mileni Versuti Ritter et al. conducted a study which suggested that rs5522 polymorphism may play a role in cardiac remodeling and aldosterone levels in drug-resistant hypertension patients with target-organ damage [23]. However, this study did not examine LV remodeling in patients post-AMI.

MRs are present in many cell types within the myocardium, including cardiomyocytes, macrophages and the coronary vasculature [5]. During AMI, glucocorticoids in the heart bind to MRs with a 10- to 30-fold higher affinity than to GRs. This can be explained by MRs having high affinity for both mineralocorticoids and glucocorticoids. Glucocorticoids typically circulate at levels 100-fold higher than those of mineralocorticoids, but MRs are likely to be constitutively occupied by glucocorticoids even at daily nadir levels. Iqbal et al. proved that there is normally low to none 11β-HSD2 activity in the heart, which converts cortisol to MR-inactive cortisone [24]. MR signaling can also be non-genomic, where MR signaling occurs rapidly within minutes of the receptor activation and does not require the nuclear translocation of the receptor–hormone complex or protein synthesis. This signaling can also be genomic, where MR signaling occurs slowly, within 10 min to a few hours after the receptor activation in the nucleus [5]. However, the main drawback of targeting these molecules is when equal (if not worse) cardiotoxic actions of the cardiac MRs, activated by aldosterone, cortisol or no specific ligand (e.g., oxidative stress, hyperkalemia), are left unopposed [24]. Fraccarolo et al. demonstrated that, in the absence of MRs, cardiomyocyte-restricted MR-knockout mice showed improved cardiac healing, along with the prevention of adverse remodeling, cardiac hypertrophy, contractile dysfunction and maladaptive gene expression post-myocardial infarction [24].

Accumulating evidence has led us to establish that, during the earliest 24 to 48 h post-AMI, life-threatening ventricular tachyarrhythmias (VT and VF/VFL) are quite common due to cellular changes at the transmural level [25]. Additionally, according to the literature, the incidence of AF in post-AMI patients varies between 6 and 21% [26]. Several studies have shown that AF is associated with poor outcomes in post-AMI patients. Although this association is not fully understood, it is hypothesized that thrombo-embolic events are the cornerstone of these adverse events. Epidemiologic and demographic analysis revealed that the majority of these patients are older, hypertensive and suffering from post-AMI heart failure, which leads to the increased rate of poorer outcomes. In a similar manner, life-threatening ventricular arrhythmias can arise due to pre-existing atrial arrhythmias (such as AF or atrial flutter), which may lead to a higher rate of events of sudden cardiac death [27,28,29].

We have established that more than 90% of patients with the rs5522 TT genotype had AF during the first day of admission post-AMI. Moreover, the rs5522 C allele protects against rhythm disorders in the early phase of AMI. Michel F. Rossier explains the physiological mechanism linking the positive chronotropic response induced by aldosterone observed in vitro with isolated ventricular cardiomyocytes, and the increased risk of ventricular arrhythmias reported in vivo in hyperaldosteronism [7]. He describes the molecular steps involved between the activation of MRs and the acceleration of spontaneous myocyte contractions, including the expression of a specific microRNA (miR204), the down-regulation of a silencing transcription factor (NRSF) and the re-expression of a fetal gene encoding a low-threshold voltage-gated calcium channel (CaV3.2). Finally, he provides evidence suggesting aldosterone-independent and redox-sensitive mechanisms of MR activation in cardiac myocytes.

The stimulation of MRs causes several modifications in the cardiac electrical activity, leading to the increased prevalence of cardiac arrhythmias. These modifications are likely due to aldosterone, which plays a role in cardiac tissue fibrosis and scar tissue formation. Similarly, MR signaling in cardiomyocytes may promote arrhythmias due to its crucial role in regulating cellular calcium homeostasis and action potential duration, as well as through adjusting calcium transients and sarcoplasmic reticulum diastolic calcium release. The triggering of both aldosterone and/or MRs may induce microvascular coronary dysfunction, which is a major risk factor in developing cardiac arrhythmias [7]. All the previously mentioned mechanisms highlight the preventative and therapeutic effect of MR antagonists in reducing the probability of severe arrhythmias that can lead to sudden cardiac death. These findings were clearly established through the Randomized Aldactone Evaluation Study (RALES) and the Eplerenone Post-Acute Myocardial Infarction Heart Failure Efficacy and Survival Study (EPHESUS) clinical trials [30,31].

Conduction defects complicating acute myocardial infarction are associated with increased morbidity and mortality [17]. There are no studies about conduction disorders after AMI and *NR3C2* gene polymorphisms. We have established that *NR3C2* rs5522, rs2070950 and rs4635799 are not associated with HAVB during the AMI hospitalization period.

## 5. Conclusions

Our study shows that post-AMI patients carrying the *NR3C2* rs2070950 GG genotype and the rs4635799 TT genotype are more likely to have better left ventricular function at 6 months. Conversely, patients with the rs5522 T allele are more likely to experience AF and VA events.

## 6. Limitations

A total of 301 consecutive AMI patients matching the inclusion criteria were initially screened. Out of them, 174 patients were excluded from the FLVR assessment due to sub-optimal echocardiographic imaging and the inability to follow the patient during the six-month period due to the loss of contact with the study participant, withdrawal from the study, COVID-19 follow-up restrictions, repeated cardiovascular events (repeat MI) or death. A total of 127 patients remained available for the FLVR assessment. All our patients were of the Caucasian race; therefore, genetic findings might not be fully extendable to other races.

## Figures and Tables

**Figure 1 ijerph-20-00012-f001:**
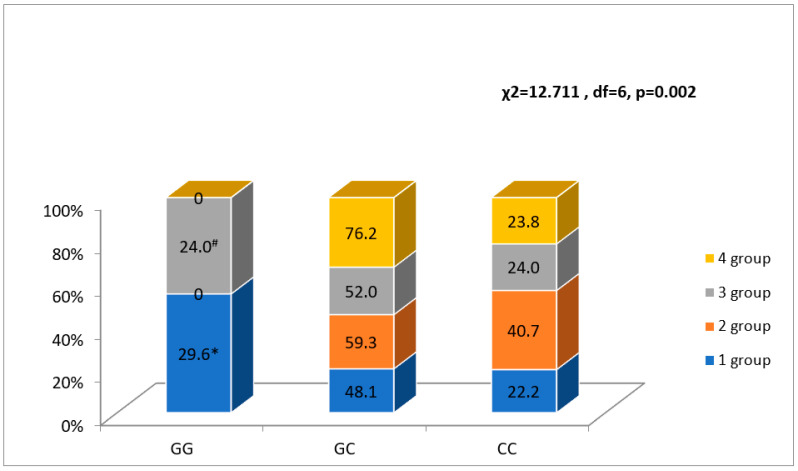
Association between NR3C2 gene polymorphism rs2070950 and FLVR after MI groups. * *p* < 0.05 as compared with groups 2 and 4, # *p* < 0.05 as compared with group 4.

**Figure 2 ijerph-20-00012-f002:**
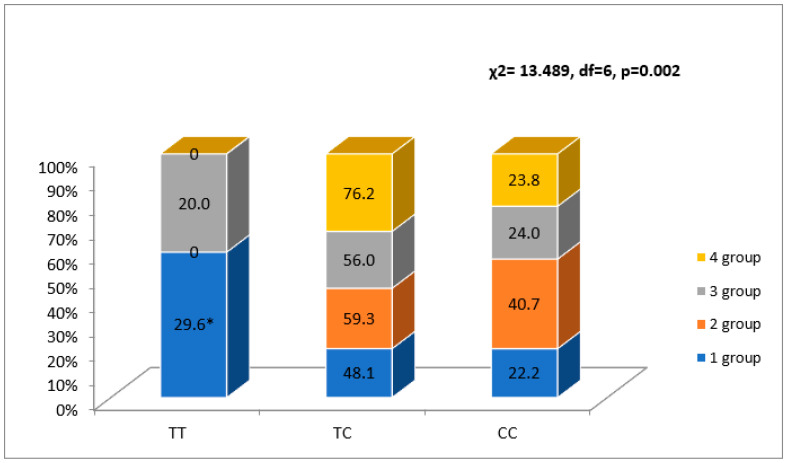
Association between NR3C2 gene polymorphism rs4635799 and FLVR after MI groups. * *p* < 0.05 as compared with group 2.

**Figure 3 ijerph-20-00012-f003:**
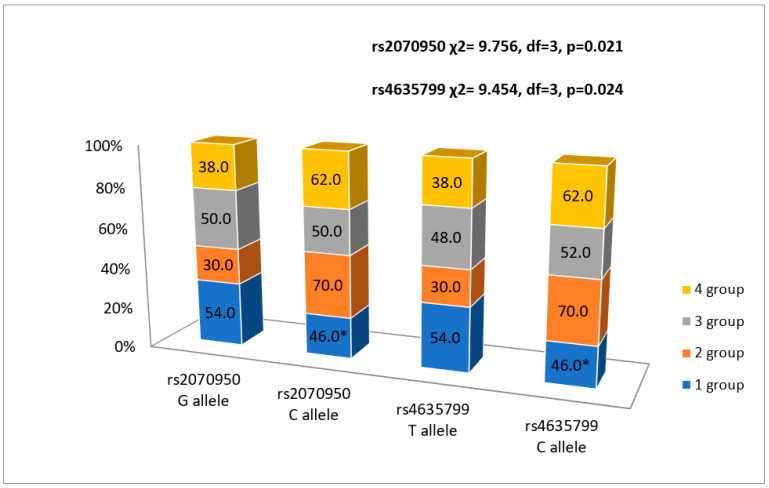
Association between rs2070950 and rs4635799 alleles and FLVR after MI groups. * *p* < 0.05 as compared with group 2.

**Table 1 ijerph-20-00012-t001:** Baseline characteristics of patients (n = 127) in the FLVR groups after AMI.

Baseline Characteristics	Group 1 n= 54	Group 2 n = 27	Group 3 n = 25	Group 4 n = 21	*p*-Value
Male n, (%)	36 (66.7)	23 (85.2)	17 (68.0)	15 (71.4)	NS
Diabetes mellitus n, (%)	7 (13.0)	1 (3.7)	5 (20.0)	3 (14.3)	NS
Hypertension n, (%)	51 (94.4)	25 (92.6)	23 (92.0)	17 (81.0)	NS
Dyslipidemia n, (%)	51 (94.4)	25 (92.6)	23 (92.0)	19 (90.5)	NS
Family history of heart attacks n, (%)	2 (3.7)	3 (11.1)	5 (20.0)	4 (19.0)	NS
Smoking n, (%)	13 (24.2) ^a^	13 (48.1) ^a,b^	6 (24.3) ^a^	14 (67.3) ^b^	0.002
Previous stroke n, (%)	4 (7.4)	2 (7.4)	2 (8.0)	5 (23.8)	NS
STEMI/NSTEMI n, (%)	43 (79.6)/11 (20.4)	24 (88.9)/3 (11.1)	20 (80.0)/5 (20.0)	19 (90.5)/2 (9.5)	NS
Dual/triple antiplatelet therapy n, (%)	51 (94.4)/3 (5.6)	27 (100)/0 (0)	22 (88.0)/3 (12.0)	19 (90.5)/2 (9.5)	NS
ACE inhibitors or ARB n, (%)	50 (92.6)	24 (88.9)	21 (84.0)	20 (95.2)	NS
Beta-blockers n, (%)	48 (88.9)	24 (88.9)	20 (80.0)	20 (95.2)	NS
Statins n, (%)	53 (98.1)	27 (100)	25 (100)	21 (100)	NS
MRA n, (%)	4 (7.4) ^a^	8 (29.6) ^b,c^	4 (16.7) ^a,c^	11 (55.0) ^b^	0.001
Age, years M (SD)	65.0 (9.4) ^a^	58.4 (10.3) ^b^	64.6 (11.1) ^a,b^	57.1 (12.3) ^b^	0.006
Troponin, μg/L median [IQR]	0.56 [0.07–7.3]	0.52 [0.06–5.1]	0.9 [0.08–10.4]	0.9 [0.21–8.1]	NS
Hbg, g/L mean (SD)	143.0 (13.9)	147.0 (20.7)	143.4 (13.5)	138 (13.7)	NS
WBC, 10^9^/L median [IQR]	9.3 [8.4–12.7]	8.7 [7.1–11.1]	8.8 [7.3–10.0]	10.8 [7.7–13.9]	NS
CRP, g/L median [IQR]	5.0 [5.0–19.3]	5.0 [2.2 –10.4]	5.0 [4.8–27.7]	6.2 [4.2–36.0]	NS
Total cholesterol, mmol/L mean (SD)	5.3 (1.1)	5.4 (1.3)	5.5 (1.1)	5.3 (1.0)	NS
HDL, mmol/L median [IQR]	1.2 [1.0–1.5]	1.2 [0.97–1.5]	1.3 [1.1–1.5]	1.1 [0.96–1.3]	NS
LDL, mmol/L median [IQR]	3.5 (1.0)	3.6 (1.1)	3.7 (0.9)	3.5 (0.9)	NS
TG, mmol/L median [IQR]	1.2 [0.8–1.7]	1.3 [0.9–1.6]	1.2 [0.9–1.5]	1.4 [1.0–2.1]	NS
Cortisol, nmol/L median [IQR]	726 [490–1019]	599.5 [423.8–832.7]	561.4 [406.6–880.2]	580.4 [368.8–1359.6]	NS
Glucose, mmol/L median [IQR]	6.8 [6.0–8.7]	5.7 [5.3–7.0]	6.3 [5.2–7.7]	6.9 [6.2–9.1]	0.028

FLVR—functional left ventricular remodeling; M—Mean; SD—standard deviation; n—number; STEMI—ST-segment elevation myocardial infarction; NSTEMI—non-ST segment elevation myocardial infarction; Hbg—hemoglobin; WBC—white blood cell; CRP—C-reactive protein; HDL—high density lipoprotein; LDL—low density lipoprotein; TG—triglyceride; ACE—angiotensin-converting enzyme; ARB—angiotensin receptor blockers; MRA—mineralocorticoid receptor antagonists; NS—not statistically significant, *p* > 0.05. Group 1—EDV < 20% and the same or increased LVEF; Group 2—EDV < 20% and a decrease in LVEF; Group 3—EDV ≥ 20% and the same or increased LVEF; Group 4—EDV ≥ 20% and decreased LVEF; ^a,b,c^—*p* < 0.05.

**Table 2 ijerph-20-00012-t002:** Echocardiography characteristics of patients in the FLVR groups after AMI.

Echocardiography Characteristics	Group 1 n= 54	Group 2 n = 27	Group 3 n = 25	Group 4 n = 21	Unadjusted OR	*p*-Value
LVEDV_1_, mL, median [IQR]	88.0 [77.9–103.6]	112.8 [91.1–135.4]	77.3 [61.2–94.4]	88.3 [70.5–108.5]	0.996 [0.985–1.007]	NS
LVEDV_2_, mL, median [IQR]	83.3 [66.2–97.0]	101.2 [83.9–123.1]	111.5 [93.0–132.6]	132.1 [104.6–170.6]	1.038 [1.026–1.051]	0.001
LVESV_1_, mL, median [IQR]	54.8 [46.1–62.4]	57.5 [46.7–68.0]	45.7 [36.9–61.0]	51.2 [30.2–67.6]	0.985 [0.967–1.003]	NS
LVESV_2_, mL, median [IQR]	39.9 [31.0–51.1]	55.5 [49.2–66.4]	58.0 [45.5–75.8]	84.5 [48.5–103.3]	1.056 [1.038–1.074]	0.001
Global longitudinal strain_1_, median [IQR]	−12.9 [−15.3–(−10.1)]	−15.7 [−17.9–(−11.9)]	−13.0 [−15.3–(−11.3)]	−13.2 [−16.3–(−9.3)]	0.978 [0.909–1.053]	NS
LVEF_1_, %, median [IQR]	41.0 [33.8–45.1]	48.6 [44.2–53.5]	40.4 [34.4 –46.7]	47.7 [40.2–55.0]	1.044 [1.005–1.084]	0.025
LVEF_2_, %, median [IQR]	50.1 [45.0–56.7]	42.9 [37.7–46.7]	48.3 [41.3–52.8]	39.9 [32.4–46.3]	0.921 [0.886–0.958]	0.001
LV SWT_1_, mm, median [IQR]	12 [11–13]	11.5 [10.7–12.0]	12.00 [11.00–14.3]	11.0 [10.0–12.8]	0.968 [0.810–1.157]	NS
LV PWT_1_, mm, median [IQR]	11 [10–12]	11 [10–12]	11.0 [10.0–12.0]	10.0 [10.0–11.1]	0.986 [0.761–1.277]	NS
LV MM_1_, median [IQR]	186.0 [167.9–227.2]	213.1 [173.9–226.8]	206.7 [187.0–253.5]	199.8 [158.0–232.8]	1.007 [1.000–1.013]	0.051
MMI_1_, median [IQR]	98.8 [86.0–111.8]	103.0 [92.0–109.4]	109.6 [98.8–118.9]	104.7 [84.0–120.9]	1.011 [0.996–1.026]	NS
Relative wall thickness_1_, median [IQR]	0.46 [0.43–0.49]	0.43 [0.39–0.47]	0.44 [0.39–0.52]	0.42 [0.38–0.51]	1.003 [0.929–1.082]	NS
Mitral E velocity_1_ (cm/s), median [IQR]	66 [56–82]	60.0 [50.5–77.0]	58.5 [48.8–86.3]	58.0 [55.0–79.0]	0.992 [0.975–1.010]	NS
Mitral A velocity_1_ (cm/s), median [IQR]	76 [63–90]	72.0 [51.0–86.0]	71.5 [55.5–83.5]	67.0 [55.9–78.0]	0.983 [0.967–0.999]	0.038
Mitral E/A ratio_1_, median [IQR]	0.79 [0.66–1.18]	0.76 [0.63–1.42]	0.87 [0.60–1.45]	0.86 [0.74–1.29]	1.376 [0.644–2.939]	NS
LV S’ average_1_, cm/s, median [IQR]	8.2 [6.7–9.4]	9.0 [7.5–11.6]	8.4 [7.0–9.9]	7.7 [6.6–9.6]	1.044 [0.902–1.209]	NS
LV E’ average_1_, cm/s, median [IQR]	6.6 [5.5–7.6]	7.8 [7.0–9.3]	6.8 [5.7–8.6]	6.7 [6.1–8.3]	1.231 [1.019–1.487]	0.031
MV anulus motion average_1_, mm, median [IQR]	11.1 [9.0–12.3]	11.6 [10.2–12.6]	10.6 [8.7–12.7]	10.0 [9.2–11.6]	0.997 [0.849–1.172]	NS
TV annulus motion_1_, mm, [IQR]	23 [18–25]	24 [20–25]	21 [19–25]	22 [17–25]	0.992 [0.928–1.061]	NS
RV S’_1_, mm, [IQR]	13.0 [11.5–15.0]	13.7 [11.8–15.0]	12.4 [11.1–15.0]	13.0 [10.7–14.2]	0.929 [0.835–1.034]	NS
LA diameter_1_, mm, [IQR]	40 [37–44]	40 [36.0–42.0]	41 [38–44]	38 [35–42]	0.986 [0.931–1.045]	NS

LV—left ventricular; EDV—end-diastolic volume; ESV—end-systolic volume; EF—ejection fraction; SWT—septal wall thickness; PWT—posterior wall thickness; MM—myocardial mass; MMI—myocardial mass index; MV—mitral valve; TV—tricuspid valve; RV—right ventricular; LA—left atrium: _1_—during the first 72 h of hospitalization for AMI; _2_—after 6-month follow-up.

**Table 3 ijerph-20-00012-t003:** Serum cortisol level association with the rhythm and conduction disorders.

Rhythm and Conduction Disorders	Serum Cortisol, nmol/L	*p*-Value
VA/no VA	1028 (324)/718 (365)	0.001
AF/AFL/no AF/AFL	981 (351)/700 (359)	0.001
HAVB/no HAVB	856 (389)/728 (367)	NS

VA—ventricular arrhythmia (ventricular tachycardia, ventricular fibrillation and ventricular flutter); AF—atrial fibrillation; AFL—atrial flutter; HAVB—high-grade atrioventricular block; NS—not statistically significant, *p* > 0.05.

**Table 4 ijerph-20-00012-t004:** NR3C2 Gene polymorphisms (n = 301) and their association with rhythm and conduction disorders.

*NR3C2* Gene Polymorphisms		Rhythm and Conduction Disorders					
		VA/No VA n = 19/282	*p*-Value	AF/AFL/No AF/AFL n = 40/261	*p*-Value	HAVB/No HAVB n = 23/278	*p*-Value
rs2070950, n (%)	GG	2 (10.5)/46 (16.3)	NS	9 (22.5)/39 (15.0)	NS	0 (0.0)/48 (17.3)	NS
	GC	10 (52.5)/148 (52.6)		18 (45.0)/140 (53.5)		15 (65.2)/143 (51.4)	
	CC	7 (36.8)/88 (31.2)		13 (32.5)/82 (31.5)		8 (34.8)/87 (31.3)	
	G allele	14 (37.0)/240 (43.0)	NS	36 (45.0)/218 (42.0)	NS	15 (33.0)/239 (43.0)	NS
	C allele	24 (63.0)/324 (57.0)		44 (55.0)/304 (58.0)		31 (67.0)/317 (57.0)	
rs4635799, n (%)	TT	2 (10.5)/47 (16.7)	NS	9 (22.5)/40 (15.4)	NS	0 (0.0) ^a^/49 (17.6) ^b^	NS
	TC	10 (52.6)/147 (52.1)		18 (45.0)/139 (53.1)		15 (65.2)/142 (51.1)	
	CC	7 (36.8)/88 (31.2)		13 (32.5)/82 (31.5)		8 (34.8)/87 (31.3)	
	T allele	14 (37.0)/241 (43.0)	NS	36 (45.0)/219 (42.0)	NS	15 (33.0)/240 (43.0)	NS
	C allele	24 (63.0)/323 (57.0)		44 (55.0)/303 (58.0)		31 (67.0)/316 (57.0)	
rs5522, n (%)	TT	19 (100)/223 (79.1)	NS	39 (97.5) ^a^/203 (77.7) ^b^	0.014	21 (91.3)/221 (79.5)	NS
	TC	0 (0.0)/54 (19.1)		1 (2.5) ^a^/53 (20.4) ^b^		2 (8.7)/52 (18.7)	
	CC	0 (0.0)/5 (1.8)		0 (0.0)/5 (1.9)		0 (0.0)/5 (1.8)	
	T allele	38 (100.0)/500 (89.0)	0.028	79 (99.0)/459 (88.0)	0.003	44 (96.0)/494 (89.0)	NS
	C allele	0 (0) ^a^/64 (11.0) ^b^		1 (1.0) ^a^/63 (12.0) ^b^		2 (4.0)/62 (11.0)	

VA—ventricular arrhythmia (ventricular tachycardia, ventricular fibrillation and ventricular flutter); AF—atrial fibrillation; AFL—atrial flutter; HAVB—high-grade atrioventricular block; NS—not statistically significant, *p* > 0.05 ^a,b^, *p* < 0.05.

## Data Availability

Not applicable.

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
