# Peer review of "Impact of Mineralocorticoid Receptor Gene NR3C2 on the Prediction of Functional Classification of Left Ventricular Remodeling and Arrhythmia after Acute Myocardial Infarction"

_ijerph, 2022, doi:10.3390/ijerph20010012_

Round 1

Reviewer 1 Report

The purpose of the study was to investigate the impact of the NR3C2 gene polymorphisms, on left ventricular remodeling, rhythm, and conduction disorders in a cohort of acute myocardial infarction patients who underwent primary percutaneous coronary intervention. The manuscript is well written, though it is not easy to interpret and understand data tables. The manuscript would benefit if data were presented in a more visual way.

Author Response

Dear Sir or Madam,

Kindly thank you for Your comments.

We attached our responses.

Kind regards,

Rima Braukyliene

Reviewer 2 Report

The manuscript is well written. Please in the conclusions write down " our study shows" instead of established.

Author Response

(The authors gave the same response as above.)

Reviewer 3 Report

Summary

In This manuscript, Rima et al. have investigated the impact of the NR3C2 gene polymorphisms on left ventricular remodeling, rhythm, and conduction disorders in a cohort of AMI patients who underwent primary percutaneous coronary intervention (PCI). As the authors mentioned that their work is the first to establish the impact of NR3C2 polymorphism on AMI, their work is significant. However, the manuscript requires considerable improvements, as mentioned in the comments.

Comments:

1.       The authors mentioned that there were more smokers in the fourth group than the others. What were the implications. The authors should include a brief outcome in their analysis.

2.       As more than 80% of the patients under study were on anti-platelet therapies, what was the outcome of these drugs? Did the authors consider this factor?

3.       The study comprised 301 patients; however, the baseline characteristic represents only 127 patients. Please explain why the remaining patients have been excluded.

4.       Table no 3 needs formatting as the polymorphism names have been cropped on the left corner, and it is impossible to identify the gene polymorphisms listed.

5.       The authors described that majority of the patients in the first FLVR group carry the rs2070950 GG genotype. At the same time, it is true that the second and fourth groups don’t have the GG genotype. It wouldn’t be correct to say that majority is GG. The table shows that GC is the major genotype in the first group for the rs2070950. Please confirm.

6.       The results described are confusing as the analysis shows that most of the FLVR 1 carries the G allele. But the authors have not mentioned this finding throughout the manuscript.

7.       The statement, "In addition, patients after MI more frequently had the rs2070950 C allele in the fourth FLVR group, is contradictory to the table. Here, group, I shows more C alleles than 4. Please confirm.

8.       Further, in the case of rs4635799, the table displayed a higher frequency in 1 than 2.

9.       Table 4 can be clearer if the NR3C2 genotypes are also included in the respective places.

10.    A piece of evidence showing the expression of the NR3C2 in AMI will be helpful. For instances RT PCR of NR3C2 in the genomic DNA isolated DNA.

11.    If within the scope of this work, the expression level of ventricular remodeling genes may help correlate the results.

12.    In section 4. The authors mentioned ventricular Flutter but have not described it throughout the manuscript. Please explain, and it appears to be a typical error.

13.    The rs5522 TT genotype frequency was higher in patients with AF/AFL than in patients without AF/AFL . Please explain the effect of the T allele, as it was noted to be higher.

14.    The first sentence of the discussion section does not reflect this paper's results. For instance, out of the three polymorphisms, only two (rs2070950, and rs4635799) were not linked to the conduction system. Also, rs5522tt was not related to the FLVR after the MI.

15.    In the references, reference number 30 is missing the paper’s title.

Author Response

(The authors gave the same response as above.)

Round 2

Reviewer 3 Report

Thank you for addressing all the comments and responding to each comments adequately. The manuscript is significantly improved.

Minor comment:

Point 12: In section 4. The authors mentioned ventricular Flutter but have not described it throughout the manuscript. Please explain, and it appears to be a typical error. Here the author’s response is still not adequate. The VFL are missing in the table 3. Please check.

Author Response

Dear Sir or Madam,

We have made some minor changes in our manuscript according to Yours comment (we have explained the term ventricular arrhythmia under table 3). You can see the changes in the attached document.

Best regards,

Rima Braukyliene
